# Investigation of the Correlation between Initial Microstructure and Critical Current Density of Nb-46.5 wt%Ti Superconducting Material

In Yong Moon [ID], Se-Jong Kim, Ho Won Lee [ID], Jaimyun Jung, Young-Seok Oh * and Seong-Hoon Kang * [ID]

Korea Institute of Materials Science, 797 Changwondaero, Seongsan-gu, Changwon-si 51508, Gyeongnam-do, Korea; mooniy085@kims.re.kr (I.Y.M.); ksj1009@kims.re.kr (S.-J.K.); h.lee@kims.re.kr (H.W.L.); jjm0475@kims.re.kr (J.J.)
* Correspondence: oostone@kims.re.kr (Y.-S.O.); kangsh@kims.re.kr (S.-H.K.)

**Abstract:** We have investigated the effect of initial microstructures on the change in critical current density (Jc) of Nb-46.5 wt%Ti (NbTi) superconducting material. It is well known that $\alpha$-Ti phases distributed in NbTi material act as a flux pinning center, resulting in an improvement in critical current density. Therefore, it is crucial to obtain the grain-refined microstructure, which is strongly related with precipitation of uniformly distributed fine $\alpha$-Ti phases and higher volume faction of $\alpha$-Ti phases, as $\alpha$-Ti phases are precipitated at the grain boundaries and triple points during heat treatments. Therefore, in order to characterize the effect of initial microstructure of NbTi on critical current density, different initial microstructures were obtained by applying equal channel angular pressing (ECAP) and hot rolling with different strains. It was revealed experimentally that hot rolling with a higher strain is efficient for obtaining the initial microstructure, which has equiaxed fine grains of $\beta$-NbTi with the aid of dynamic recrystallization, and which is helpful for precipitating fine $\alpha$-Ti phases during intermediate heat treatment. Furthermore, it was confirmed that critical current density can be enhanced by obtaining a smaller $\alpha$-Ti phase, a higher aspect ratio of $\alpha$-Ti phase, a higher volume fraction of $\alpha$-Ti phase and a ribbon-like folded $\alpha$-Ti phase.

**Keywords:** Nb-46.5 wt%Ti; superconducting; microstructure; dynamic recrystallization



## 1. Introduction

With recent development in medical technology, diseases in the brain, liver and muscles can be rapidly and accurately diagnosed using magnetic resonance imaging (MRI). As high-resolution images play a decisive role in disease diagnosis, interest in high-field MRI has increased. Making a magnet with a high magnetic field is one of the key points for high magnetic field MRI. Therefore, it is essential to increase the superconducting properties of the conductive wire, which is the main component of the magnet.

Generally, NbTi is the most widely used material for MRI magnets. In addition, NbTi wire is used in cryomagnetic systems, particle accelerators (Large Hadron Collider (LHC)) and large engineering research projects (poloidal field coils in international thermonuclear experimental reactor (ITER)), etc. [1–5]. When the NbTi wire is exposed to an external magnetic field, the Lorentz force acts simultaneously with current flow, and the vortex formed inside the NbTi material becomes unstable, resulting in the vortex moving and deteriorating the sustainability of the superconducting effect. However, if a large amount of homogeneously distributed fine $\alpha$-Ti phases emerges inside the NbTi material, $\alpha$-Ti phases can serve as a flux-pinning center that keeps the vortex stable, thereby improving critical current density ($J_c$) [6–9]. Therefore, it is important to optimize the NbTi wire fabrication process to precipitate as many $\alpha$-Ti phases as possible. Generally, it is well known that $\alpha$-Ti phases are precipitated at grain boundaries and triple points through repetitive plastic deformation and intermediate heat treatments.

Based on this background, various studies have been conducted to improve the critical current density of the NbTi superconducting wire by changing its initial microstructure [10–12]. Beloshenko et al. [13,14] presented an enhancement of superconducting properties using equal channel multi-angle pressing (ECMAP). By combining ECMAP and low temperature drawing, critical current density was enhanced by about 20%. Sun et al. [15] revealed that plastic deformation at high temperatures makes high-angle grain boundaries in the NbTi material. However, the relationship between high-angle grain boundaries and superconducting properties was not analyzed in this research. Despite various studies, research on the correlation between the microstructure and critical current density, using specimens in which initial strain is induced at high process temperatures, is still insufficient.

Recently, our research group reported on a paper that analyzed the effect of initial strain level of NbTi billet on superconducting properties using equal channel angular pressing (ECAP) [16]. It was revealed that superconducting properties can be improved by applying severe plastic deformation in the NbTi billet. In addition, results on how the initial temperature of NbTi materials affects superconducting properties were reported [17]. However, the effect on superconducting properties considering both initial process temperature and initial strain level was not analyzed due to the difference of process parameters in both studies.

In this study, NbTi billet was deformed until the accumulated strain reached 2.0 by applying repeated hot rolling at a process temperature of 600 °C to compare results to our previous studies [16,17]. Thereafter, groove rolling, drawing and intermediate heat treatments were applied to make the final NbTi monowire. Subsequently, the microstructure and critical current density of the final monowire were measured, and the effect of the initial process conditions on the superconducting properties was systematically summarized by comparative analysis with previous research. As a result, it has been found that grain refinement and grain size homogenization of the initial NbTi material are important factors in improving superconducting properties. Furthermore, it was confirmed that critical current density could be enhanced by obtaining a higher volume fraction of $\alpha$-Ti phase, a smaller $\alpha$-Ti phase, a higher aspect ratio of $\alpha$-Ti phase and ribbon-like folded $\alpha$-Ti phase.

## 2. Materials and Methods

### 2.1. Materials

NbTi material, which is generally used as a low-temperature superconducting material, was applied. Table 1 shows an energy dispersive spectroscopy (EDS) result presenting chemical composition. Before 1st drawing, the NbTi specimen was assembled with Cu and Nb tubes. After assembly, the cross-section of monowire had a complex structure in which the NbTi rod were wrapped with the Nb and Cu tubes, as shown in Figure 1 (view 'B'). The Nb tube served as a diffusion barrier which prevented the formation of intermetallic compound between Cu and NbTi. The Cu tubes acted as paths for a current flow in the case of a rapid change from a super-conducting state to a non-superconducting state, as well as thermal and electrical stabilization layers [16].

**Table 1.** Chemical composition of NbTi material measured by energy dispersive spectroscopy (EDS).

| Element | Nb | Ti | C | O |
|---------|------|------|-----|-----|
| Content (wt%) | 49.6 | 46.5 | 1.7 | 2.2 |

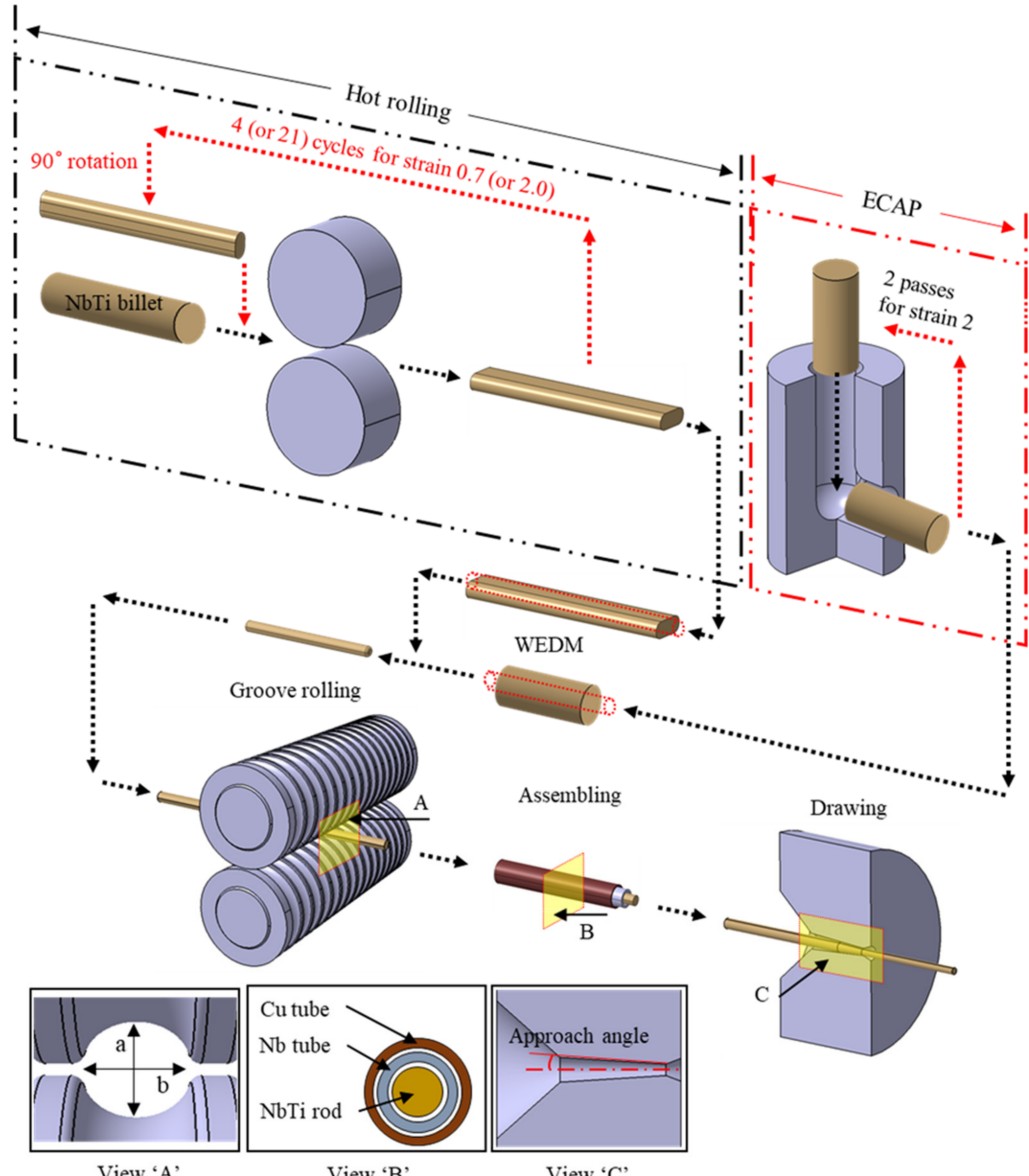

**Figure 1.** Schematics showing the whole process, include hot rolling, equal channel angular pressing (ECAP), groove rolling and drawing. Hot rolling and ECAP were conducted at 600 °C and room temperature, respectively.

### 2.2. Processes for Plastic Deformation

Figure 1 shows the overall process concept for making the superconducting NbTi monowire from the initial NbTi billet. For ease of comparison, the ECAP process used in our previous study is shown together [16]. In ECAP, if the specimen is pressed after being inserted into the top inlet of the ECAP mold, it undergoes a high level of shear deformation by passing through a channel with a 90° bending angle [18–25]. The diameter of the billet was not changed after this process. The ECAP mold was designed to apply strain of 1.0 per pass. Therefore, in this paper, the results of an ECAPed specimen subjected to 2 passes (strain 2.0) were compared with the results of the hot rolled specimen.

In order to induce a strain of 2.0 into NbTi billet in the hot rolling process, the initial NbTi billet was processed for 21 cycles at an initial process temperature of 600 °C. The strain ($\varepsilon_e$) induced in NbTi billet in the hot rolling was calculated as

$$\varepsilon_e = \frac{1}{3}\left[2\left(\left(\varepsilon_x - \varepsilon_y\right)^2 + \left(\varepsilon_y - \varepsilon_z\right)^2 + \left(\varepsilon_z - \varepsilon_x\right)^2\right)\right]^{\frac{1}{2}} \tag{1}$$

where $\varepsilon_x$, $\varepsilon_y$, and $\varepsilon_z$ represent the strain in the X, Y, and Z directions, respectively.

After the hot rolling, Ø15.0 mm rod-shaped specimens were produced by wire electrical discharge machining (WEDM), and the 1st heat treatment was performed at 420 °C for 80 h, so that the α-Ti could be precipitated at grain boundaries and triple points of the NbTi specimen. In general, it is well known that performing intermediate heat treatment when the accumulated strain reaches about 2.0 is the most effective method for α-Ti precipitation. Therefore, intermediate heat treatments were performed based on the accumulated strain of 2.0 between wire manufacturing processes. During heat treatment, annealing effects also could be induced to the NbTi specimen [26–31].

The heat treated NbTi specimen was subjected to groove rolling. The groove shape of roll was prepared as an oval, with a ratio (b/a) of 1.03, as shown in Figure 1 (view 'A'). In order to make the cross section of the NbTi specimen circular, each groove rolling was performed via rotating the NbTi specimen 90°. The NbTi specimen was deformed from Ø15.0 mm to Ø5.2 mm (the assigned strain was approximately 2.12) by groove rolling. Thereafter, a 2nd heat treatment was performed under the same conditions as the 1st heat treatment. azz Prior to performing the 1st drawing, the NbTi specimen was assembled with the Cu tube (outer diameter of 9 mm, 0.7t) and Nb tube (outer diameter of 7 mm, 0.7t). In order to remove impurities, including the oxide layer on the material surfaces, the Cu tube was pickled using nitric acid. The Nb tube and NbTi specimen were subjected to pickling treatment using hydrofluoric acid for about 30 min. After assembly, the monowire was subjected to the 1st drawing using a drawing mold with an approach angle of 12°. The diameter of the monowire was changed from 9.0 mm to 3.5 mm (the assigned strain was approximately 1.89). Here, the monowire indicates the specimen after assembly. The drawn monowire was subjected to a 3rd heat treatment under the same conditions as the 1st and 2nd heat treatments. After that, the heat-treated monowire was given a total strain of 2.51 through 2nd drawing to make a final diameter of 1.2 mm. The equipment and experimental results are shown in Figure 2.

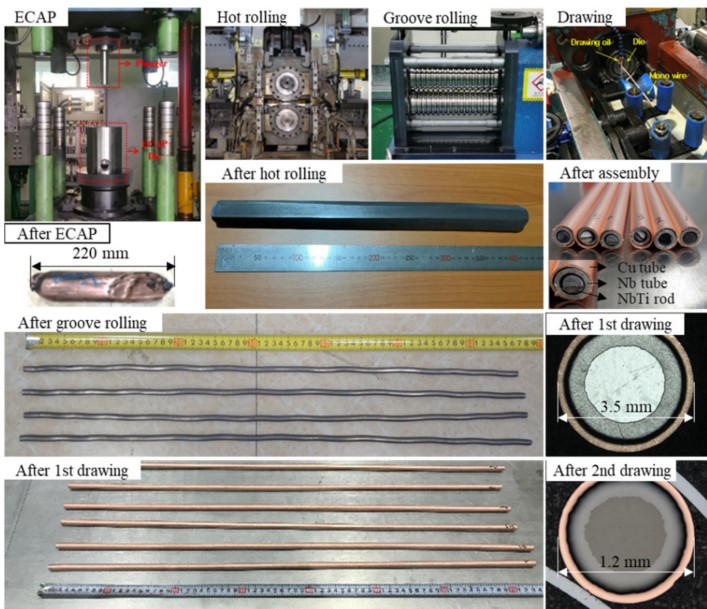

**Figure 2.** Photos depicting the various equipment for ECAP, hot rolling, groove rolling and drawing, and experimental results showing the prepared specimen and its cross-sections.

Table 2 summarizes the overall process sequence, and the strain subjected to the specimen in each sequence. As can be seen, the subsequent processes of hot rolling with strain 2.0 were carried out in the same condition as the ECAP. The strains ($\varepsilon$) applied in

the groove rolling and drawing processes presented in the table were calculated using the following equation.

$$\varepsilon = \ln \frac{A_o}{A_d} \qquad (2)$$

Here, $A_o$ and $A_d$ represent the initial cross-section area and deformed area, respectively.

**Table 2.** Experimental procedure of deformation processes and heat treatments.

| Sequence | | 1 | 2 | 3 | 4 | 5 | 6 | 7 |
|---|---|---|---|---|---|---|---|---|
| | Process | Groove Rolling | 1st Heat Treatment | Groove Rolling | 2nd Heat Treatment | 1st Drawing | 3rd Heat Treatment | 2nd Drawing |
| Hot rolling (strain 2.0) or ECAP (strain 2.0) [16] | Diameter change | × | - | 1.5 mm → 5.2 mm | - | 9.0 mm → 3.5 mm | - | 3.5 mm → 1.2 mm |
| | Accumulated strain | × | - | ≒ 2.12 | - | ≒ 1.89 | - | ≒ 2.51 |
| | Specimen condition | NbTi rod | ← | ← | ← | Monowire (Cu tube + Nb tube + NbTi rod) | | |
| Hot rolling (strain 0.7) [17] | Diameter change | 20 mm → 15 mm | - | 15 mm → 8.4 mm | - | 17 mm → 6.4 mm | - | 6.4 mm → 1.0 mm |
| | Accumulated strain | ≒ 0.58 | - | ≒ 1.15 | - | ≒ 1.95 | - | ≒ 3.66 |
| | Specimen condition | NbTi rod | ← | ← | ← | Monowire (Cu tube + Nb tube + NbTi rod) | | |

### 2.3. Material Characterization

An optical microscope (OM, Olympus, Tokyo, Japan) was used to observe the microstructure, according to processes. The specimens to be observed were mounted using carbon powder, and then mechanically polished in the order of sandpaper, diamond abrasive and colloidal suspension. Thereafter, etching was performed for about 10 s using a solution of hydrofluoric acid, nitric acid and water, in a ratio of 2:1:7, respectively.

To observe α-Ti phases, the back scatter electronic (BSE) function of a field emission scanning electron microscope (FE-SEM, TESCAN Co., Brno, Czech Republic) was applied. The area and shape of α-Ti phases on the BSE images were computationally measured using an image analysis program (Leopard-iXM, ZOOTOS Co. 2.5, Uiwang-si, Republic of Korea), and the averaged values were used for quantitative evaluation. The microstructure evolution was analyzed using the electron backscattered diffraction (EBSD). At a magnification of ×10,000, the specimen surface was scanned with a step size of 40 nm. When the misorientation angle between adjacent grains was 5° or more, they were considered to be different grains in the grain size analysis. Specimen preparation was the same for the BSE specimen.

In order to check the superconducting properties of the NbTi monowire, a physical property measurement system (PPMS, Quantum Design Co., San Diego, CA, USA) was used. The measurement was performed by changing the magnetic field from −10 T to +10 T. The atmosphere temperature was fixed at 4.2 K using liquid helium. The NbTi monowire with a length of 5 mm was used for the PPMS measurement. The magnetic moment ($M$) was measured and converted to critical current density ($J_c$) using Bean's model [32]. The calculation formula for the critical current density is as follows.

$$J_c = 30 \times \frac{\Delta M}{d_{equ}} \qquad (3)$$

Here, $d_{equ}$ represents the equivalent diameter of NbTi without Nb and Cu tubes.

### 3. Results

### 3.1. Analysis of OM Images

Figure 3 shows the microstructure of the NbTi before and after initial processes (ECAP and hot rolling). The microstructure of the ECAPed specimen was taken in the radial

direction (RD), while the microstructures of the hot rolled specimen were taken in the transverse direction (TD), as indicated at the bottom-left side of Figure 3.

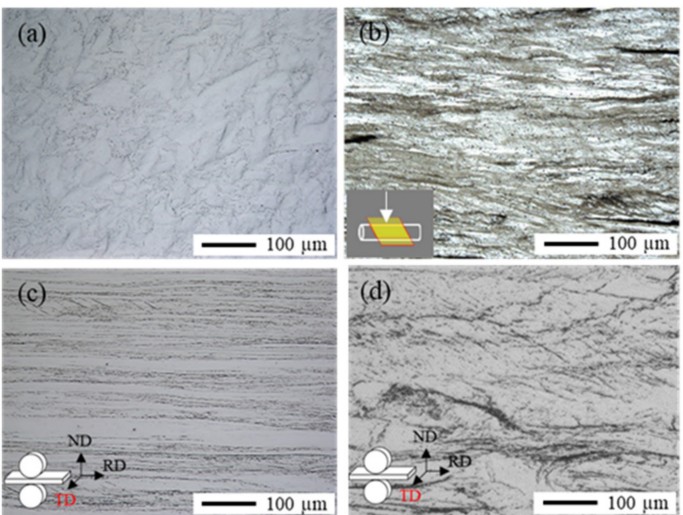

**Figure 3.** Microstructures obtained from (**a**) as-received NbTi billet, (**b**) ECAPed specimen with strain 2.0 [16], (**c**) hot rolled specimen with strain 0.7 [17] and (**d**) hot rolled specimen with strain 2.0.

As shown in Figure 3a, the as-received specimen has coarse β-NbTi grains. In the case of the ECAPed specimen (Figure 3b), the β-NbTi grains were severely deformed, so that it is difficult to observe the microstructure clearly. Otherwise, the hot rolled specimen with strain 0.7 shows a clear band structure formed in the rolling direction (Figure 3c). However, when the induced stain reaches 2.0, the band structure is blurred due to entangled grain boundaries (Figure 3d). In particular, the ECAPed specimen appears to have more complex grain boundaries than the hot rolled specimen with strain 2.0. This is thought to be due to its different process temperature and mode of deformation. Therefore, in the next section, we will analyze how this difference, according to the process conditions, affects α-Ti precipitation and superconducting properties.

### 3.2. Effect of Initial Process Conditions on α-Ti Precipitation

3.2.1. Qualitative Analysis Using BSE Images

Figure 4 shows the measured BSE images of each process step from the first heat treatment to the second drawing. The dark areas in the figures indicate α-Ti phases. As shown in Figure 4a, spherical α-Ti phases were precipitated after the first heat treatment under all initial process conditions. In the case of the ECAPed specimen, both large and small size α-Ti phases exist together. In contrast, relatively uniform size α-Ti phases are distributed at the grain boundaries and triple points of the hot rolled specimen. The volume fraction of α-Ti phases tends to increase as the strain increases from 0.7 to 2.0. This is because the stored energy increases with increasing plastic strain, thereby promoting the nucleation of α-Ti phases in the subsequent heat treatment processes [33].

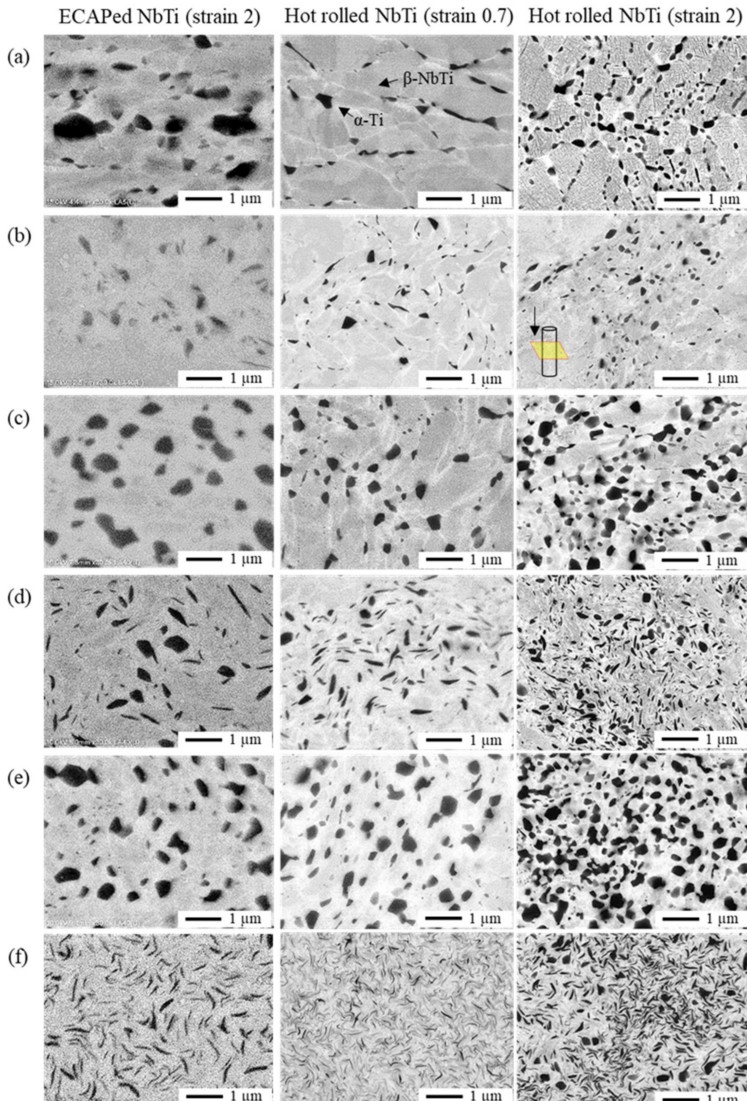

**Figure 4.** BSE (back scatter electronic) images depicting the microstructures after (**a**) 1st heat treatment, (**b**) groove rolling, (**c**) 2nd heat treatment, (**d**) 1st drawing, (**e**) 3rd heat treatment and (**f**) 2nd drawing. All images are taken from an axial direction. BSE images of ECAPed specimen with strain 2.0, and hot rolled specimen with strain 0.7, are reused from references [16,17], respectively.

Figure 4b shows that some of the spherical α-Ti phases are partially flattened due to compressive deformation during the first groove rolling. However, it seems that they did not receive sufficient deformation (or compressive stress) to change all spherical α-Ti phases to flat α-Ti phases. Subsequently, as the second heat treatment proceeds, the partially flattened α-Ti phases disappear, and all α-Ti phases change to a spherical shape (Figure 4c). This is because the intermediate heat treatments spheroidized the flat α-Ti phases and also precipitated new α-Ti phases at grain boundaries and triple junctions [30]. The peculiar thing is that, in the case of ECAPed specimen, the size of α-Ti phases after the second heat treatment is smaller compared to the first heat treatment step, whereas the size of α-Ti phases in the hot rolled specimen after the second heat treatment seems to have increased compared to the first heat treatment. It is believed that this is because the abnormally large α-Ti phases were precipitated in the ECAPed specimen during the first heat treatment.

As the first drawing and third heat treatment proceed, the shape of the α-Ti phases is transformed into a needle shape (flattened shape); it then changes to a spherical shape again (Figure 4d,e). Figure 4f shows the morphology of the α-Ti phases after the second

drawing. Most of the $\alpha$-Ti phases are transformed into a folded or curved needle shape after the second drawing due to a high level of plastic strain. In addition, it appears that the volume fraction of $\alpha$-Ti phases increases remarkably compared with the microstructure of the first heat treatment step. In conclusion, the $\alpha$-Ti phases were effectively refined, and the volume fraction of the $\alpha$-Ti phases increased through repetitive plastic deformation and heat treatments.

Figure 5 shows the shape of $\alpha$-Ti phases measured in the radial direction after the second drawing. It can be seen that $\alpha$-Ti phases are elongated in the axial direction due to the repeated groove rolling and drawing processes. Elongated $\alpha$-Ti phases have a positive effect on the improvement of superconducting properties because they allow the magnetic flux lines, which have penetrated into the superconductor, to flow continuously and stably in the longitudinal direction of the wire [7]. Therefore, it can be concluded that the applied plastic deformation processes effectively developed the NbTi microstructure so as to improve superconducting properties.

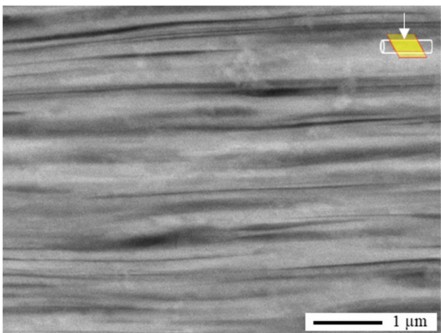

**Figure 5.** BSE image depicting the microstructure of the ECAPed specimen after 2nd drawing measured in the radial direction.

3.2.2. Quantitative Analysis

Figure 6 presents the change of equivalent diameter, aspect ratio and volume fraction of $\alpha$-Ti phases, according to the process steps. In the case of the ECAPed specimen, the equivalent diameter of the $\alpha$-Ti phases after the first heat treatment is relatively larger than those of the hot rolled specimens (Figure 6a). Additionally, the size of the $\alpha$-Ti phases becomes smaller after the second heat treatment. On the contrary, in the hot rolled specimens, the size of the $\alpha$-Ti phases after the first heat treatment is relatively small. The equivalent diameter increases after the second heat treatment. This difference tendency seems to be due to the large $\alpha$-Ti phases precipitated in the first heat treatment in the ECAPed specimen, as shown in Figure 4a. After the 3rd heat treatment, a similar reduction level in the equivalent diameter occurs in all initial process conditions.

The average aspect ratio tends to increase as cold work progresses (or applied strain in cold works, such as groove rolling and drawing, increases). This is because the spherical $\alpha$-Ti phases are deformed into a thin needle shape by plastic deformation during the subsequent cold works. It is noteworthy that the increase rate of average aspect ratio in ECAPed specimen is relatively small compared to that of the hot rolled specimen. It indicates that, even under the same plastic strain, largely precipitated $\alpha$-Ti phases (in the ECAPed specimen) are difficult to change into flat or folded $\alpha$-Ti phases with a high aspect ratio. The aspect ratio of the hot rolled specimen with a strain of 0.7 is higher than those of the ECAPed and hot rolled specimens with a strain of 2.0, due to the higher strain level applied in the second drawing (the applied strain in the second drawing was 3.66 for the hot rolled specimen with a strain of 0.7, and 2.51 for the ECAPed specimen/hot rolled specimen with a strain of 2.0).

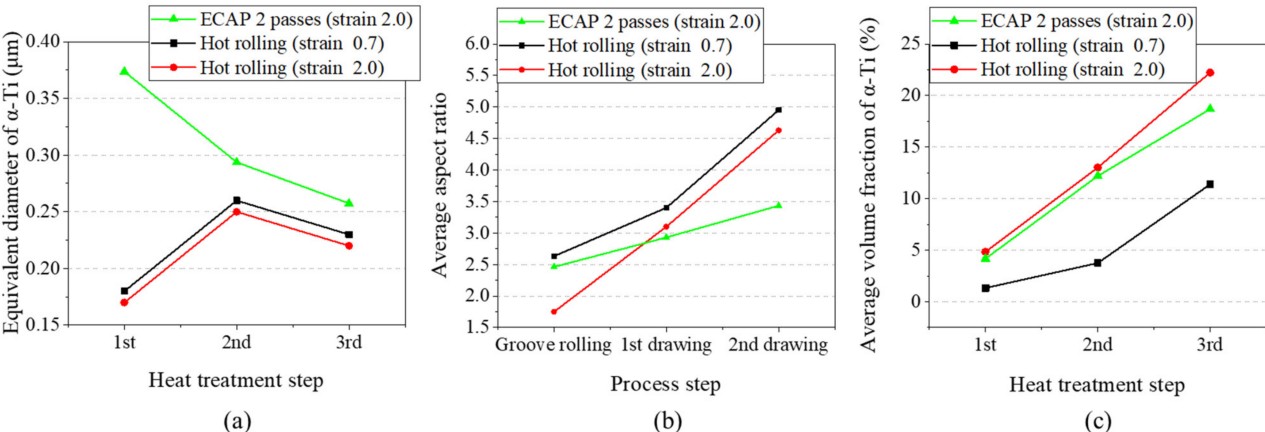

**Figure 6.** Graphs depicting (**a**) equivalent diameter of the α-Ti phase after each heat treatment, (**b**) average aspect ratio of the α-Ti phase after each cold work and (**c**) average volume fraction of the α-Ti phase after each heat treatment. Results of the ECAPed specimen with a strain of 2.0 and the hot rolled specimen with a strain of 0.7 are reused from references [16] and [17], respectively.

Figure 6c presents the average volume fraction of the α-Ti phases. In all initial process conditions, the volume fraction of α-Ti phases increases as heat treatments proceeds. This is evidence by the fact that the volume fraction of a α-Ti phase effectively increased through repeated cold works and heat treatments. It is expected that the volume fraction of α-Ti can be increased by applying more steps of the process. However, the wire diameter reached 1.2 mm, so further steps cannot be applied. The volume fraction of the α-Ti phases is the smallest in the hot rolled specimen with a strain of 0.7 in all heat treatment steps, and is enhanced in the ECAPed specimen and the hot rolled specimen with a strain of 2.0.

Figure 7a shows the normalized moment values of the NbTi monowire after the second drawing. The moment value measured between the magnetic field (−0.5 T)–(+0.5 T) is unstable. Therefore, the data from 0 T to 0.5 T were excluded from the critical current density calculation. The critical current density of the wire-shaped specimen is obtained by substituting the Δ*M* of Figure 7a into Equation (2), and the results are presented in Figure 7b. In all cases, critical current density increased and reached a peak point as the magnetic field decreased. After passing its peak point, critical current density continued to decrease. The specimen subjected to hot rolling with a strain of 2.0 showed the highest critical current density in the whole magnetic field range, and the maximum value was measured at around 1.2 T. Particularly, this maximum value of the critical current density was 3.2 times larger than that of the ECAPed specimen. The ECAPed specimen showed the lowest critical current density. This is considered to be due to a relatively large equivalent diameter and a low aspect ratio of the α-Ti phases compared to the hot rolled specimens. Additionally, the number of α-Ti phases folded like a ribbon shape appears to be less compared to the hot rolled specimens.

### 3.3. Microstructure Analysis Using EBSD Measurement

Figure 8 shows the grain distributions before the first heat treatment, according to the initial process conditions. As shown in Figure 8a,b, the as-received specimen has coarse β-NbTi grains. When ECAP is performed twice to induce a strain of 2.0, a grain refinement partially occurs (Figure 8c,d). In contrast, relatively uniform grain refinement occurs in the hot rolled specimens with the strains of 0.7 and 2.0 (Figure 8e–h). This difference is more pronounced after the first heat treatment.

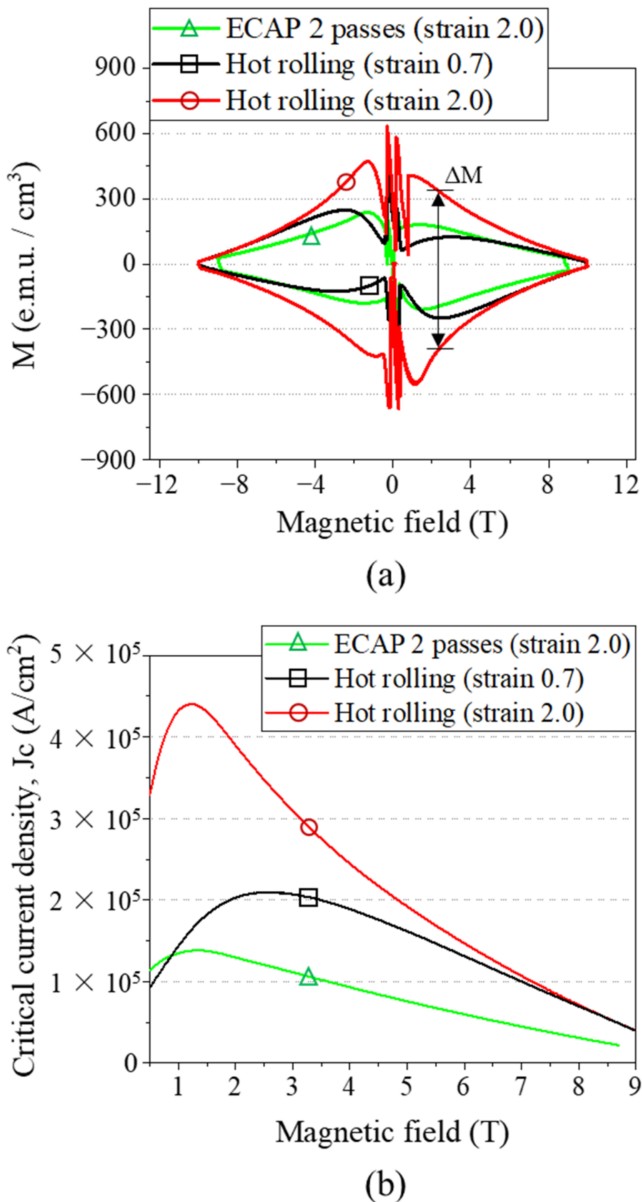

**Figure 7.** (**a**) Normalized moment according to initial process conditions and (**b**) calculated critical current density.

In the case of the ECAPed specimen, the size of the refined grains appears to increase after the first heat treatment (Figure 9a,b). There is no clear evidence for static recrystalliza­tion in the ECAPed specimen during the first heat treatment. It seems that grain growth occurred in the existing refined grains during heat treatment. In addition, in the part where grain refinement did not occur during ECAP, no significant change was observed, even after heat treatment. On the other hand, in the hot rolled specimen with a strain of 2.0, numerous small grains were newly formed after the first heat treatment (Figure 9e,f). These fine grains seem to be formed by static recrystallization. That is, the high dislocation density accumulated by hot rolling acts as a driving force to generate static recrystallization during the first heat treatment. With respect to the formation of sub-grains, the ECAPed specimen rarely had sub-grains inside unrefined grains (Figure 9b). However, in the case of the hot rolled specimen with a strain of 2.0, lots of sub-grains (red and green lines) exist inside the grain, with a high angle boundary (blue lines, area 'A' in Figure 9f). Here, the red, green, and blue lines indicate the misorientation angle of 2–5°, 5–15° and over 15°, respectively. Therefore, it is expected that, through further cold works and heat treatment,

small α-Ti phases can be precipitated evenly in the fine β-NbTi grain boundaries and their triple points.

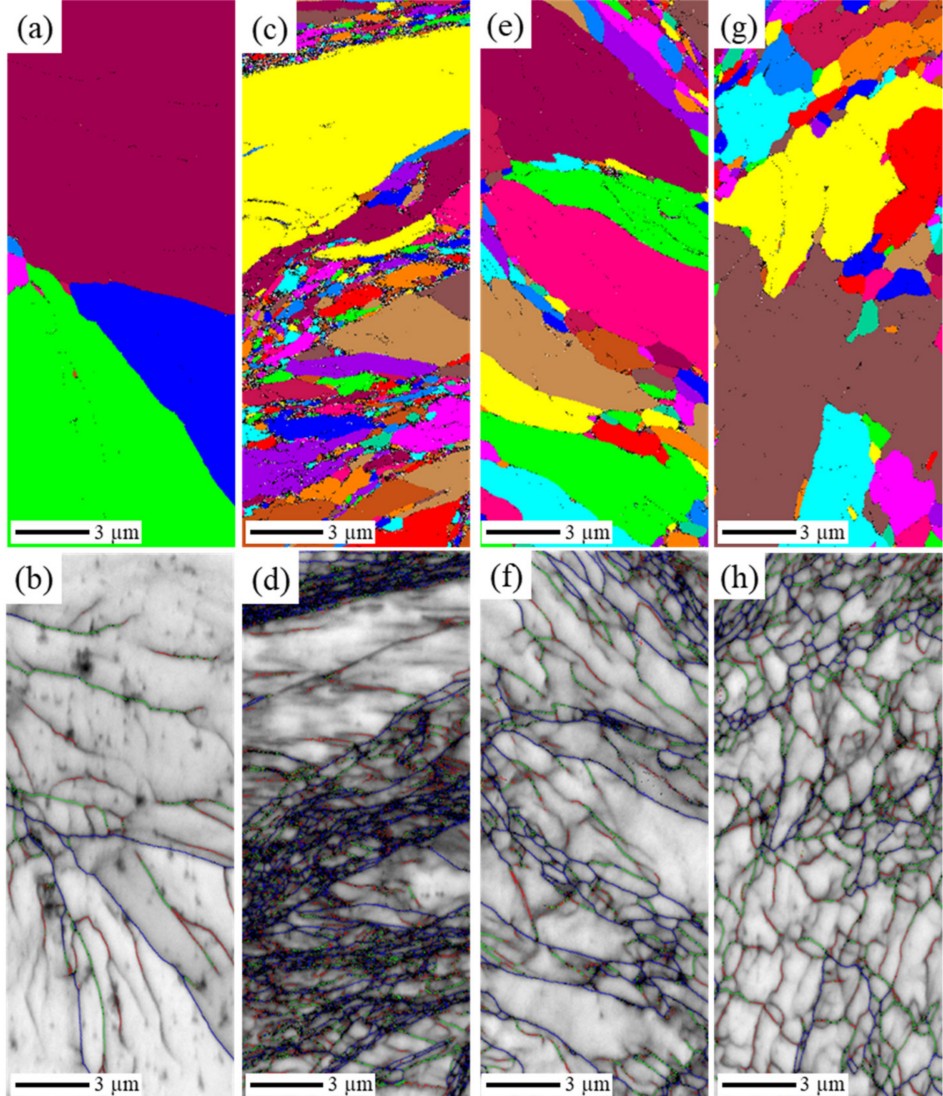

**Figure 8.** EBSD images depicting grain distributions before 1st heat treatment, measured from (**a**,**b**) as-received NbTi billet, (**c**,**d**) the ECAPed specimen with a strain of 2.0, (**e**,**f**) hot rolled specimen with a strain of 0.7 and (**g**,**h**) hot rolled specimen with a strain of 2.0. Red, green, and blue grain boundaries indicate the misorientation angle of 2–5°, 5–15° and over 15°, respectively.

The microstructure evolution before and after the first heat treatment can be clearly confirmed in Figure 10. In the case of the ECAPed specimen, the number of β-NbTi grains with a size of 0.25 μm decreased from 940 to 158 after first heat treatment. On the other hand, the number of β-NbTi grains with a size of 0.25 μm in the hot rolled specimen with a strain of 2.0 tended to increase from 83 to 419 after first heat treatment. This supports the above results that each initial process condition has a great influence on recrystallization behavior.

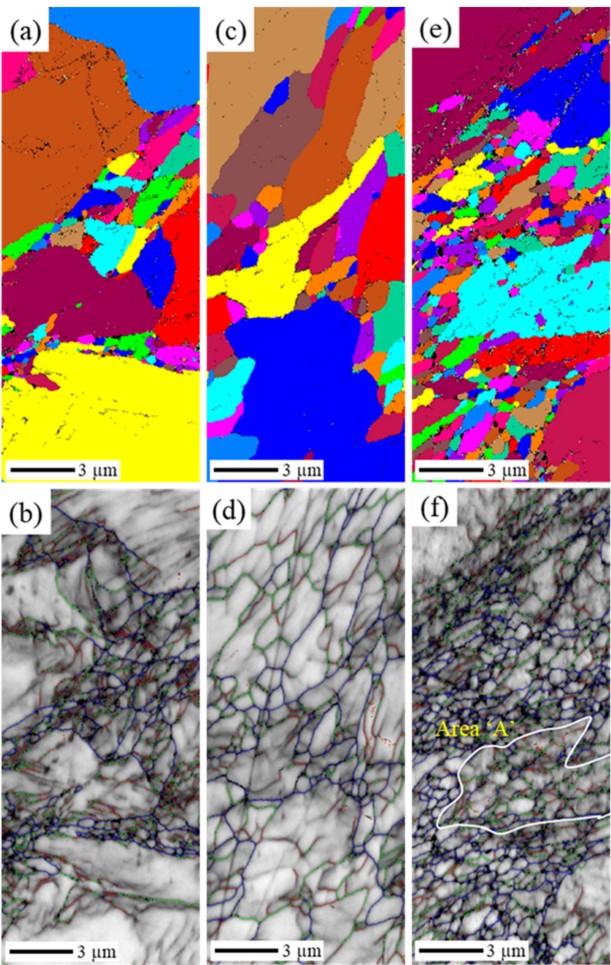

**Figure 9.** EBSD images depicting grain distributions after 1st heat treatment, measured from (**a**,**b**) the ECAPed specimen with a strain of 2.0, (**c**,**d**) the hot rolled specimen with a strain of 0.7, and (**e**,**f**) the hot rolled specimen with a strain of 2.0. Red, green, and blue grain boundaries indicate the misorientation angle of 2–5°, 5–15° and over 15°, respectively.

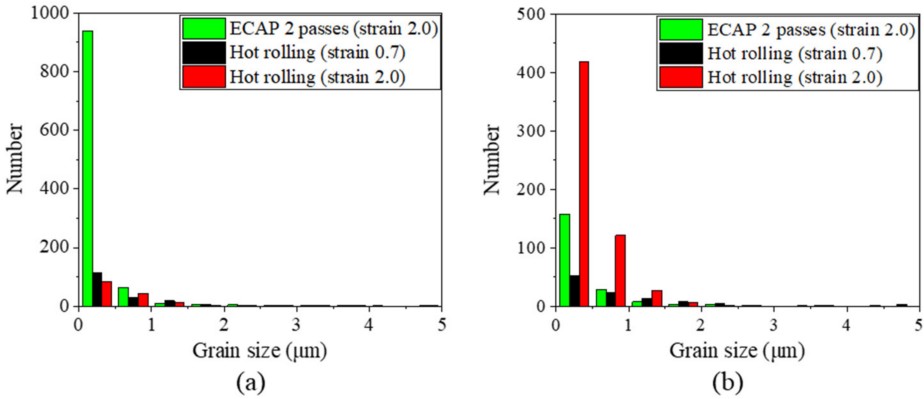

**Figure 10.** Grain size distribution of β-NbTi according to the initial process conditions (**a**) before and (**b**) after 1st heat treatment.

## 4. Discussion

Figure 11a shows the schematics of the β-NbTi microstructures before and after the first heat treatment. As described in Section 3.3, when hot rolling was performed to induce a strain of 2.0, grain refinement occurred relatively uniformly. However, in the case of the

ECAPed specimen with a strain of 2.0, refined and non-refined grains exist simultaneously. This difference is considered to have a great influence on the precipitation of α-Ti phases after the first heat treatment. As shown in Figure 11a, in the hot rolled specimen with a strain of 2.0, the grain boundary is evenly distributed over the entire area due to static recrystallization after the first heat treatment. This means that the grain boundary and triple point where α-Ti phases can be precipitated are evenly distributed over the whole region. Therefore, α-Ti phases can be precipitated uniformly in the hot rolled specimen. On the other hand, in the case of the ECAPed specimen, a non-homogeneous grain refinement characteristic is maintained after the first heat treatment, as only grain growth without static recrystallization occurred during the first heat treatment. This means that large α-Ti phases can be precipitated near coarse grains, and small α-Ti phases can be precipitated in the vicinity of fine grains, as shown in Figure 11a [34]. Therefore, it is posited that the abnormally large α-Ti phases in Figure 4a were formed by the abovementioned reason.

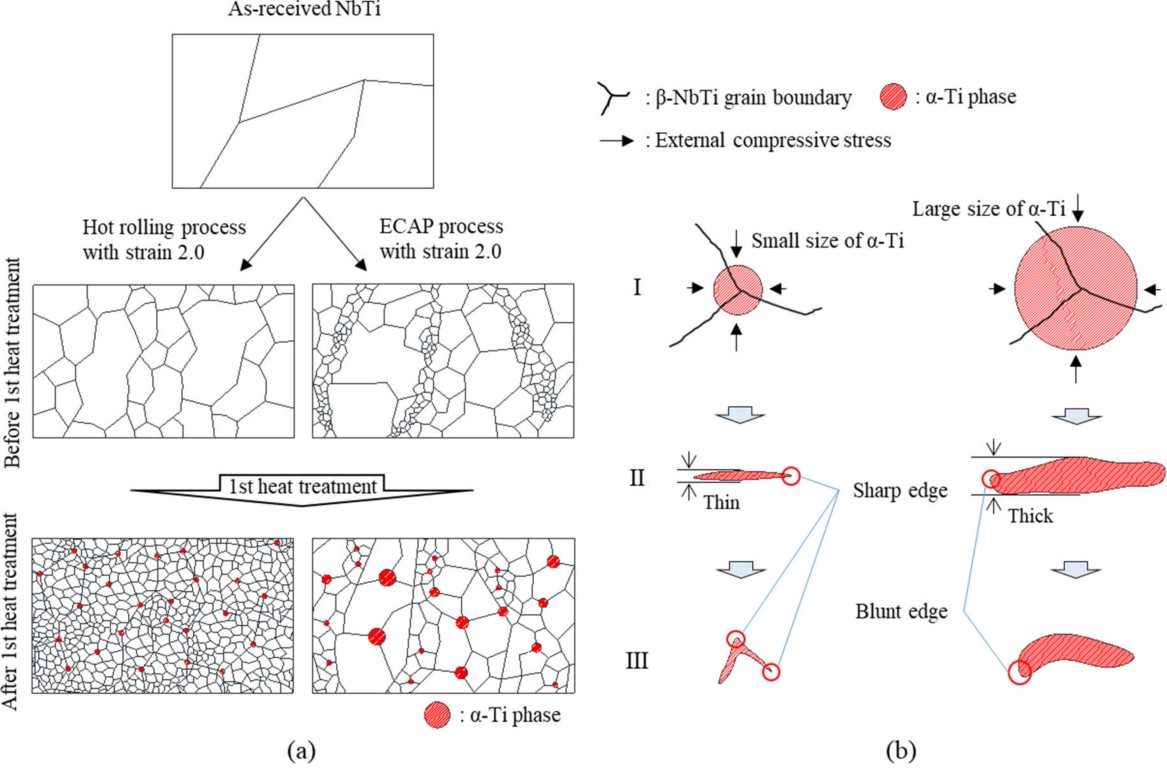

**Figure 11.** Schematics depicting (**a**) β-NbTi microstructures according to process sequence and (**b**) deformation progress of small and large α-Ti phases.

Figure 11b shows the deformation progress of the small and large α-Ti phase. During cold works (groove rolling and drawing), α-Ti phases are subjected to anisotropic stress, depending on the grain orientation of surrounding β-NbTi grains (I). Therefore, the α-Ti phase compressed by the anisotropic stress is first deformed into a flat shape (II). As can be seen in Figure 11, the relatively small α-Ti phase has sharper edges compared to the larger one. In addition, a small and flat α-Ti phase is thinner than a large and flat α-Ti phase (II). Therefore, the transformation from a flat to a folded shape might easily occur in the small and flat α-Ti phase (III).

The shape of α-Ti phases affects superconducting properties. That is, when the aspect ratio of the α-Ti phases increases and the thickness of flat α-Ti phases decreases, the edges of a flat α-Ti phase act as a stronger pinning center. This characteristic is related to coherence length, which is 5 nm for NbTi. Li et al. [7] suggest that the optimum superconducting property appears when the thickness of the α-Ti phase is similar to the coherence length of NbTi. Meingast et al. [35] reported that, when the thickness of the α-Ti phase decreases from

100 nm to 10 nm, the pinning force increases linearly. Furthermore, if the number of folds in the ribbon is $n_f$, the folded ribbon shows the same flux pinning force as a set of ($n_f$ + 1) pieces of flat ribbons [36]. Thus, the number of pinning points in a ribbon $n_a$ is simply estimated by $n_a = n_f + 1$. Therefore, it can be inferred that the ECAPed specimen, which not only has a relatively low aspect ratio (or thicker $\alpha$-Ti phases) but also a smaller number of folded $\alpha$-Ti phases, has a disadvantage in improving superconducting properties. The analysis of superconducting characteristics through a shape analysis of $\alpha$-Ti phases shows a good agreement with critical current density measurement results (Figure 7).

Much research on severe plastic deformation processes, such as ECAP, show that the refined grains are inhomogeneous [22–25]. Increasing the number of passes in the ECAP process will alleviate inhomogeneity from a macroscopic point of view. However, microscopically, inhomogeneity still exists. On the other hand, it has been reported that grain refinement through hot rolling can obtain homogeneous grains, as the microstructure is refined with dynamic recrystallization [26–28]. Therefore, it is considered that the hot rolling is more suitable for grain refinement to enhance superconducting properties.

## 5. Conclusions

In this study, we investigated the effects of initial strain level and process temperature on superconducting properties using Nb-46.5 wt%Ti material. NbTi billet was initially processed by repetitive hot rolling at 600 °C to induce a strain of 2.0. After that, an NbTi monowire with a diameter of 1.2 mm was made through groove rolling and drawing processes.

The prepared hot rolled NbTi specimen with a strain of 2.0 was analyzed through comparisons with our previous research, including the ECAPed specimen with a strain of 2.0 and a hot rolled specimen with a strain of 0.7. As a result, it was confirmed that the specimen subjected to hot rolling had better superconducting properties than the ECAPed specimen, regardless of the induced strain level.

To analyze the cause of the above results, microstructure was measured through EBSD. It was found that grain refinement occurred unevenly in the case of the ECAPed specimen. Therefore, the precipitation of the $\alpha$-Ti phases occurred non-uniformly in the subsequent processes, resulting in deteriorating superconducting properties. It was revealed that hot rolling with the higher strain is quite efficient for obtaining an initial microstructure with equiaxed fine grains of $\beta$-NbTi, with the aid of dynamic recrystallization, which is helpful for precipitating fine $\alpha$-Ti phases during intermediate heat treatment. Furthermore, it was confirmed that critical current density can be enhanced by obtaining the smaller $\alpha$-Ti phase, the higher aspect ratio of $\alpha$-Ti phase, the higher volume fraction of $\alpha$-Ti phase and ribbon-like folded $\alpha$-Ti phase.

**Author Contributions:** Conceptualization, I.Y.M. and S.-H.K.; methodology, H.W.L.; validation, Y.-S.O., J.J. and S.-H.K.; formal analysis, S.-J.K.; investigation, I.Y.M.; resources, H.W.L.; data curation, J.J.; writing—original draft preparation, I.Y.M.; writing—review and editing, Y.-S.O.; visualization, J.J.; supervision, S.-H.K.; project administration, Y.-S.O.; funding acquisition, S.-J.K. All authors have read and agreed to the published version of the manuscript.

**Funding:** This work was supported by the Technology Innovation Program (N0002598 and N10053590) funded by the Ministry of Trade, Industry & Energy (MOTIE, Korea).

**Institutional Review Board Statement:** Not applicable.

**Informed Consent Statement:** Not applicable.

**Data Availability Statement:** The data presented in this study are available on request from the corresponding author. The data are not publicly available due to the Korea Institute of Materials Science (KIMS) data management regulations.

**Conflicts of Interest:** The authors declare no conflict of interest.

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
