# Peer review of "Investigation of the Correlation between Initial Microstructure and Critical Current Density of Nb-46.5 wt%Ti Superconducting Material"

_metals, doi:10.3390/met11050777_

Round 1

Reviewer 1 Report

Article reference: metals-1210096. Title: Investigation on Correlation between Initial Microstructure and Critical Current Density of Nb-46.5wt%Ti Superconducting Material by
In Yong Moon, Se-Jong Kim, Ho Won Lee, Jaimyun Jung, Young-Seok Oh, and Seong-Hoon Kang 

The manuscript reports that hot rolling with the higher strain is efficient to obtain the initial microstructure having the equiaxed fine grains of β-NbTi, which is helpful to precipitate fine α-Ti phases during intermediate heat treatment and confirmed that the critical current density can be enhanced by obtaining the higher volume fraction of α-Ti phase, the smaller α-Ti phase, the higher aspect ratio of α-Ti phase and ribbon-like folded the α-Ti phase. These findings provide valuable data to optimize the NbTi wire fabrication. Therefore, I recommend the publication of the manuscript on Metals after some issues are addressed.

  1. The organizing sequence of Fig.5 is different with abstract as volume fraction, size, and aspect ratio. It is better to be consistent.
  2. The manuscript does not give the optimal treatment to get the highest volume fraction, the smallest, and the highest aspect ratio of the α-Ti phase. Base on the result shown in Fig.5 the reviewer guess more heat treatment and process steps and higher strain will get better results. Will higher strain break the NbTi wire? If not higher strain and more heat treatment and process steps should be employed to get the optimal way of getting the best α-Ti phases level.

Author Response

First of all, we greatly thank for the reviewers’ valuable comments. We tried to give the best answers to the questions.

All modifications were highlighted using ‘Track Changes’ function in Microsoft Word.

[Reviewer 1]

Comment - The organizing sequence of Fig.5 is different with abstract as volume fraction, size, and aspect ratio. It is better to be consistent.

Answer.

  • As reviewer’s recommend, the organizing sequence in abstract and conclusion were modified [line 22~23, 399~400]

Comment - The manuscript does not give the optimal treatment to get the highest volume fraction, the smallest, and the highest aspect ratio of the α-Ti phase. Base on the result shown in Fig.5 the reviewer guesses more heat treatment and process steps and higher strain will get better results. Will higher strain break the NbTi wire? If not higher strain and more heat treatment and process steps should be employed to get the optimal way of getting the best α-Ti phases level.

Answer.

  • In general, it is well known that performing heat treatment when the accumulated strain reaches about 2.0 is the most effective for α-Ti precipitation. Therefore, we performed intermediate heat treatments when the accumulated strain reached to 2.0.

  • Additionally, when the accumulated strain reaches 2.0 or higher, excessive forming load may cause poor forming or break the wire during drawing.

  • As the reviewer comment, if the heat treatment and process steps are further conducted, superconductivity may be better. However, diameter of the wire becomes smaller than the desired shape (Ø2 mm).

  • Therefore, it is necessary to find the optimization methods for α-Ti precipitation within the given process steps, and as a result, it is proposed to hot-roll the initial specimen.

  • Related information has been added to the manuscript. [line 107 ~ 111 and 259 ~ 261]

Reviewer 2 Report

In this paper Moon and co-authors present the results of experimental studies of correlations between processing, structure and superconductive properties of NbTi wire.

NbTi is a very old superconductive system, the wires are produced commercially for decades and have  been subjected to seemingly all possible optimizations. Nevertheless the results are interesting for the technological development because they clearly highlight the way of improvement of the wire properties. The paper is well written an gives eliminating understanding of the performed research. All technological processes and structural studies are clearly described: it is clearly shown that formation of α-Ti phase as pinning centers could be promoted by hot rolling and additional heat treatment.

I recommend to publish tis paper, after a few minor issues are addressed:

  1. It is evident seen that due rolling and drawing process the structure of the wire has a certain anisotropy, that may in turn cause anisotropic pining. It should be explained therefore how the magnetic field was oriented with respect to wire.
  2. I think the authors should discuss the possible applications in the Introduction section more broadly. Besides MRI, NbTi wire is used to build laboratory cryomagnetic systems, particle accelerators (high-energy physics) and other magnetic elements for neutron physics or e.g. ITER project.
  3. Fig.6 should be discussed in more detail. 

Author Response

Responses to review

First of all, we greatly thank for the reviewers’ valuable comments. We tried to give the best answers to the questions.

All modifications were highlighted using ‘Track Changes’ function in Microsoft Word.

[Reviewer 2]

Comment - It is evident seen that due rolling and drawing process the structure of the wire has a certain anisotropy, that may in turn cause anisotropic pining. It should be explained therefore how the magnetic field was oriented with respect to wire.

Answer.

  • The α-Ti phases were elongated in the longitudinal direction by repeated cold working and became an anisotropic shape.
  • It is well known that elongated α-Ti phases has a positive effect for improving the superconducting properties.
  • These information was added in manuscript along with additional figure. [line 222 ~ 232]

Comment - I think the authors should discuss the possible applications in the Introduction section more broadly. Besides MRI, NbTi wire is used to build laboratory cryomagnetic systems, particle accelerators (high-energy physics) and other magnetic elements for neutron physics or e.g. ITER project.

Answer.

  • As reviewer’s comment, some of applications were added. [line 33 ~ 35]

Comment - Fig.6 should be discussed in more detail.

Answer.

  • As reviewer’s comment, additional information was added. [line 274 ~ 276]